# A Synchronous Magnitude Estimation with P-Wave Phases’ Detection Used in Earthquake Early Warning System

**DOI:** 10.3390/s22124534

**Published:** 2022-06-16

**Authors:** Dingwen Zhang, Jihua Fu, Zhitao Li, Linyue Wang, Jiale Li, Jianjun Wang

**Affiliations:** 1National Institute of Natural Hazards, Ministry of Emergency Management of China, Beijing 100085, China; zdw55922@163.com (D.Z.); zhitao.lee@163.com (Z.L.); 17860612108@163.com (L.W.); wjj2855@vip.sina.com (J.W.); 2Institute of Disaster Prevention, Sanhe 065201, China; jialelee666@163.com

**Keywords:** earthquake early warning system, synchronous magnitude estimation, P-wave phases’ detection, STP/LTP, SNR

## Abstract

How to estimate an earthquake’s magnitude rapidly and accurately is a challenge for any earthquake early warning system. In order to reach a balance between accuracy and timeliness, a synchronous magnitude estimation method with P-wave phases’ detection is proposed. In this method, the P-wave phases are detected by the changes of the signal-to-noise ratio (SNR) of the seismic records, where the SNRs are calculated by the short-term power and long-term power ratio (STP/LTP). Meanwhile, the variations of the SNR are applied to estimate the magnitude of the earthquake. By the statistics of some earthquake cases, a synchronous magnitude estimation model of the variation of the P-wave phases’ SNR, the earthquake magnitude, and the hypocentral distance was built. Compared with some other magnitude estimation methods, the suggested method inherits the robustness of the STP/LTP method and is more accurate and rapid than the peak displacement ($P_{d}$) method.

## 1. Introduction

A rapid and accurate earthquake warning system (EEW) can effectively reduce casualties and economic losses caused by earthquakes. The concept of an earthquake early warning system was first proposed by Dr. Cooper of the United States in 1868 [1] and has been built in many areas and achieved good results in earthquake prevention and mitigation [2,3,4,5,6]. Where magnitude estimation plays a key role in earthquake early warning system, the data used to estimate the magnitude are obtained by seismic sensors. The MEMS accelerometer in the seismic sensor monitors the ground motion data in real time, then calculates the parameters required for magnitude estimation through the corresponding algorithm. On the premise of stable performance of the seismic sensors [7], it is a challenge to improve the timeliness and accuracy of the magnitude estimation algorithm. As two important parameters for estimating the magnitude of an earthquake, timeliness and accuracy are always a pair of contradictions. A large number of research efforts have been made to keep the balance between accuracy and timeliness. At present, there are many methods for real-time magnitude estimation, mainly including periodic methods, displacement amplitude methods, energy methods, and machine learning methods.

The $\tau_{p}^{max}$ method, as one of periodic methods, was first proposed by Nakamura, which calculates the predominant period of ground motion according to real-time seismic velocity records [1]. This method has certain limitations, and its accuracy and stability are easily affected by sampling rate and data preprocessing. Selecting different filters or filtering time windows will have a great impact on the calculation results of the characteristic period. Allen and Kanamori improved the $\tau_{p}$ algorithm, and it was regarded that the maximum value of the dominant period within a few seconds of the P waves phases’ arrival is proportional to the magnitude. This method is applied to the Elarms system in the United States and has achieved good results in earthquake early warning [3]. Then, Kanamori proposed the $\tau_{c}$ method based on the predominant period $\tau_{p}^{max}$ method, which improved the calculation method of characteristic period and was more stable and reliable than the $\tau_{p}^{max}$ method [8]. However, this method shows a saturation phenomenon in estimating strong earthquakes, and its accuracy is also affected by the length of time window. Moreover, Wu et al., carried out studies on the basis of the $\tau_{c}$ method [9]. Yamada et al., found that the $\tau_{c}$ method displays the saturation phenomenon for big earthquakes [10]. In the aspect of displacement amplitude method, Wu et al., proposed the $M_{L10}$ method in 1998, which comprehensively calculates the estimation results of earthquake magnitude by using the waveform records of all successive triggering stations within 10 s of the first triggering station [11]. Unfortunately, this method is only suitable for the areas with high network density. Wu and Kanamori found that there was a good correlation between the displacement amplitude $P_{d}$ and the peak velocity PGV of ground motion within 3 s after P waves phase’s detection [12]. After that, Wu and Zhao proposed the empirical relationship between $P_{d}$, hypocentral distance *R* and magnitude *M*, which is more stable than $\tau_{p}$ method and $\tau_{c}$ method [13]. This method is more stable than the $\tau_{p}$ method and $\tau_{c}$ method, but there are still some limitations. The $P_{d}$ method also has the saturation phenomenon in the large earthquake, and with the increase of time window, the critical ‘saturation’ magnitude will increase accordingly. Lancieri et al., further studied Wu’s method with strong earthquake records with hypocentral distance within 50 km [14]. Li et al., and Wang et al., conducted a comparative analysis of the periodic methods and the displacement amplitude methods, and established the corresponding regression model [15,16].

As for the energy methods, the cumulative absolute velocity (CAV) parameter is used as one of important criteria for magnitude estimation in the EEW system in Istanbul, Turkey [17]. The limitation of this method is that the parameter calculation needs a certain time. In 2008, Yamada et al., proposed a new magnitude estimation parameter, intensity magnitude $M_{I}$. According to the recorded results of the station, the magnitude can be calculated in real time and the results can be updated [10].

In recent years, with the rapid development of computer technology, machine learning and other technologies have also been applied in magnitude estimation. For example, Zhu et al., used a deep revolutionary neural network to estimate magnitude, and Mousavi et al., and Hu et al., applied machine learning methods to achieve accurate magnitude estimation [18,19,20]. The error of the machine learning model in this method is obviously smaller than that of the traditional method, but its parameters need to be determined at the 3 s after the arrival of P waves, which is the limitation in the timeliness.

The magnitude estimation of the above-mentioned methods is carried out a few seconds after the P waves phase’s detection in EEW system. Usually, the magnitude estimation and the P waves phase’s detection are accordingly figured out. A method for synchronous magnitude estimation with P waves’s phase detection is proposed in this paper. In the proposed synchronous magnitude estimation method, one P waves phase’s detection method based on signal-to-noise ratio (SNR) was modified to estimate the magnitude right after the P phases detection [21]. The novel magnitude estimation method is put forward according to the relationship among the instantaneous maximum SNR, magnitude *M*, and hypocentral distance *R*. Based on the same algorithm model, the magnitude estimation is synchronously executed with the P waves phase’s detection. The proposed method can effectively improve the efficiency of magnitude estimation in EEW system.

## 2. Materials and Methods

At present, many studies support the magnitude estimation based on a small amount of initial information of earthquakes, and the $\tau_{c}$ method and $P_{d}$ method are widely used. There is a nucleation seismic phase in the initial fracture process of the fault, which directly determines the final fracture morphology and the magnitude of the earthquake [22]. P waves and S waves produced by the earthquake carry a lot of seismic information, P waves carry the information of fault sliding, and S waves carry the main energy information of the earthquake. In this paper, it is assumed that there is a certain weak seismic energy in P waves, and there is a certain attenuation relationship between the energy and hypocentral, and the magnitude is estimated by using this initial energy information and hypocentral.

The relationship between the magnitude *M*, maximum displacement peak $P_{d}$, and hypocentral distance *R* proposed by Wu et al., is shown in Equation (1).
(1)$$M=A\times log\left(P_{d}\right)+B\times log\left(R\right)+C$$
where *M* is the surface wave magnitude, *A*, *B*, and *C* are the fitting coefficients, $P_{d}$ is the peak amplitude of displacement in the first three seconds after the arrival of the P waves, and *R* is the hypocentral distance.

The STP/LTP method provides a new method for P waves’ arrival extraction and rapid calculation of instantaneous SNR [21]. Here, it can be considered to use the maximum ratio $P_{SNR}$ of effective energy of instantaneous SNR to background noise as a new measure to replace the $P_{d}$ parameter to reconstruct this relationship model so as to realize the synchronous estimation of earthquake magnitude. According to the literature and observation records, the regression relationship among magnitude *M*, $P_{SNR}$, and hypocentral distance *R* is put forward, namely
(2)$$M=A\times log\left(P_{SNR}\right)+B\times log\left(R\right)+C$$
where *M* is the surface wave magnitude, *A*, *B*, and *C* are the fitting coefficients, $P_{SNR}$ is the instantaneous maximum value of STP/LTP corresponding to the arrival of P waves, and *R* is the hypocentral distance.

The STA/LTA method is often used as the criterion for P waves’ arrival. The long-time window represents the energy of background noise and the short-time window represents the energy of P waves’ effective signal. When the background noise is stable, namely when the background noise conforms to the state of stationary random process, the effective signal energy of P waves can be used as the representative of seismic energy. The Equation (3) is the calculation equation of STA/LTA.
(3)$$\frac{STA_{i}}{LTA_{i}}=\frac{\sum_{j=i+1-S}^{i}CF_{j}/S}{\sum_{k=i+1-L}^{i}CF_{k}/L}$$
where $STA_{i}$ and $LTA_{i}$ are the mean value of characteristic function of short window and long window respectively, *CF* is the characteristic function, and *S* is the length of the short window and *L* is the length of the long window.

According to the internal relationship between signal-to-noise ratio (SNR) and STA/LTA, Fu et al., proposed the automatic extraction method (STP/LTP) of seismic P waves’ arrival time based on SNR. SNR is often used to represent the relative strength of effective signal and background noise. The SNR calculation equation is shown in Equations (4) and (5).
(4)$$SNR=10\mathrm{lg}\left(P_{e}/P_{n}\right)$$
(5)$$dSNR=P_{e}/P_{n}$$
where SNR is signal-to-noise ratio, $P_{e}$ is the effective signal power, $P_{n}$ is the background noise power, and dSNR is the signal-to-noise ratio expressed linearly. Where $P_{e}$ and $P_{n}$ are usually described by relative power, the calculation method is Equation (6):(6)$$P=\sum_{i=1}^{N}{\left(x_{i}-\bar{x}\right)}^{2}/N$$
where $x_{i}$(i=1,2,…,N) is the value of the observed signal and $\bar{x}$ is the mean value of the observed signal.

In the STP/LTP method, in order to represent the energy of the ground vibration signal effectively, the characteristic function is constructed from the perspective of relative power, as shown in Equation (7).
(7)$$CF={\left(x_{i}-\bar{x}\right)}^{2}$$

The STP/LTP equation is
(8)$$\frac{STP_{i}}{LTP_{i}}=\frac{\sum_{j=i+1-S}^{i}{\left(x_{j}-{\bar{x}}_{j}\right)}^{2}/S}{\sum_{k=i+1-L}^{i}{\left(x_{k}-{\bar{x}}_{k}\right)}^{2}/L}=\frac{P_{S}^{i}}{P_{L}^{i}}$$

When the STP/LTP value exceeds a certain threshold, the arrival of P waves can be determined. At this time, the peak value of STP/LTP in P waves is $P_{SNR}$, as shown in Equation (9).
(9)$$P_{SNR}=\max\left(\frac{STP_{i}}{LTP_{i}}\right)$$

Figure 1 shows the application flowchart of the new method. In the practical application scenario, the sensor performs real-time monitoring, and the data is transmitted to the computer in real time to calculate the STP/LTP at each time in the observation signal. When the STP/LTP exceeds the preset trigger threshold at a certain time, that means that the arrival of P waves is determined, which is recorded as the arrival time of P waves, and then the $P_{SNR}$ of P waves is determined. Finally, the $P_{SNR}$ and the hypocentral distance obtained are substituted into the relationship model for magnitude estimation.

## 3. Performance Analysis and Results

Sichuan Province of China is located between the Indian Ocean Plate and the Eurasian Plate, where crustal movement is very frequent. There are many seismic belts in Sichuan Province. Under the action of multiple seismic belts, the underground rock strata are fractured and misplaced, resulting in the occurrence of large and small earthquakes in the region. Wenchuan M8.0 earthquake on 12 May 2008 caused a large number of casualties and economic losses.

Data for this study were provided by the National Key Research and Development Program of China (No. 2019YFC1509205), Institute of Engineering Mechanics, China Earthquake Administration and China Earthquake Networks Center, National Earthquake Data Center (http://data.earthquake.cn, accessed on 31 October 2021). Figure 2 shows the locations and distributions of the main epicenters and stations. There are 75 records of large, medium, and small earthquakes for model fitting and performance analysis, including Wenchuan M8.0 (12 May 2008), Jiuzhaigou M7.0 (8 August 2017), Luxian M6.0 (16 September 2021), Xingwen M5.7 (16 December 2018), Changning M4.7(17 November 2021), Changning M4.6 (21 November 2021), Changning M4.0 (27 November 2019), M3.6 (17 March 2020), and M3.0 (12 December 2020, 23 March 2021) in Sichuan Province. In addition, eight earthquake events are selected for model testing, including Gongxian M3.0 (13 March 2021), Gongxian M3.4 (7 November 2020), Gongxian M3.8 (23 June 2020), Gongxian M4.0 (June 12, 2020, 15 February 2021), Gongxian M4.1 (13 November 2020), Xingwen M5.7 (16 December 2018), Luxian M6.0 (16 September 2021). In these records, the verticle (U−D) direction ground motion data has the stable background noise, which is convenient for seismic phase identification. In the process of selecting ground motion data, the principle of station data selection is as follows: for the *M* ≤ 5.0 earthquakes, the station data with hypocentral distance less than 50 km are selected; for the *M* > 5.0 earthquakes, station data with hypocentral distance less than 100 km are selected.

### 3.1. Small Earthquake Data Analysis

In the sample data, the data of 3.0≤M≤4.7 small earthquakes were provided by the National Key Research and Development Program of China (No. 2019YFC1509205). The data is monitored by low-cost MEMS seismic sensors named MEMS Network Strong Motion Seismograph (MNSMS), which is equipped with a high-performance three-axis linear Class C MEMS accelerometer, and the photograph of the MNSMS is shown in Figure 3. The MNSMS is mainly composed of some hardware modules: the MEMS accelerometer module, TCP/IP module, Power over Ethernet (PoE) module, and local storage module (optional). The MNSMS can meet the needs of dense EEW in terms of noise, dynamic range, useful resolution, reliability, and detecting capabilities with its high-performance [7]. The sampling rate of the MNSMS is 50 Hz.

Figure 4 shows the records of verticle (U-D) direction of a M4.0 earthquake event and the corresponding results of STP/LTP, STP, LTP calculation. In Figure 4c, it can be clearly seen that when the P wave arrives, the STP/LTP value increases significantly. Soon, the $P_{SNR}$ value appears, which indicates that the ground motion information contains some energy at the initial stage of the P waves, and the energy is significantly more than the energy of the background noise. It can be found that the $P_{SNR}$ value corresponding to the S wave is smaller than the P wave’s, which is not that the energy of S waves is small, but the observation signal in the long-time window is mostly the energy of seismic waves. In Figure 4e,f, when the seismic signal does not arrive, the relative power in the short-time window is the relative power of the background noise, which is close to 0, and the state is very stable. When the seismic waves arrive, the relative power in the short-time window is the sum of the relative power of the seismic signal and the relative power of the background noise in the short-time window, but the relative power of the background noise in the short-time window is much smaller compared with the relative power of the seismic signal. Therefore, in the process of an earthquake, the relative power of the short-time window can represent the energy of the ground motion signal. Finally, when the seismic waves are over, only background noise is left in the short-time window, and the relative power is back to about 0. In Figure 4g,h, when the seismic signal does not arrive, the relative power of the long-time window is the relative power of the background noise, the value is close to 0, and the state is also very stable. When the seismic waves arrive, the relative power of the long-time window is the sum of the relative power of the seismic signal in the long-time window and the relative power of the background noise. In the process of an earthquake, the relative power of the long window can also represent the energy of the ground motion signal. However, unlike the short-time window, the length of the long window is very long. After averaging, the relative power of the seismic signal increases and decreases slowly. Finally, when the seismic waves are over, only background noise is left in the long-time window, and the relative power is back to about 0.

It is worth noting that when the noise is weak and stable, to a certain extent, although long-time and short-time windows can represent the energy of seismic waves, our purpose is to estimate the magnitude through the initial information of P waves. If the short-time window is just at the arrival time of P waves, then the short-time window represents the relative power of P waves, while the relative power of P waves in the long-time window is only S/L after averaging. Since S is much less than L, the relative power of the long-time window is still considered as the power of background noise at the arrival time of P waves. In this way, when the P waves arrive, it can not only effectively detect the P waves, but also obtain the relative energy proportion at the initial time of the P waves, and the $P_{SNR}$ value appears soon after the arrival of the P waves, which is the significance of this method. Multiple experimental results show that the time of the $P_{SNR}$ value is very close to the time of the P waves’ arrival, which usually occurs within 2 s after the arrival of the P waves, and sometimes even occurs simultaneously with the time of the P waves’ arrival.

Small earthquakes occur frequently in earthquake events. The signal’s energy of small earthquakes is weak and the SNR is low, so the detection of P waves’ arrival time faces many difficulties. The STP/LTP method has strong robustness, can adapt to weak noise, general noise, and strong noise; its misjudgment rate and missing rate are low. The $P_{SNR}$ method follows the advantages of STP/LTP method and can accurately obtain the $P_{SNR}$ value in strong noise environment.

For the *M* < 5.0 small earthquakes, the energy information carried by the P waves is not very strong, and the information cannot be monitored by the stations far away. Figure 5 shows the relationship between the $P_{SNR}$ value and the hypocentral distance in a M4.0 earthquake. It can be seen that when the hypocentral distance reaches 50 km, the $P_{SNR}$ value has dropped to below 4, which makes it vulnerable to background noise. In addition, Figure 5 also reflects the attenuation relationship between $P_{SNR}$ value and hypocentral distance. With the increase in hypocentral distance, the energy of seismic waves decreases gradually, so the $P_{SNR}$ value will also decrease.

### 3.2. Large Earthquake Data Analysis

The large earthquake data (*M* = 6, 7, 8) were provided by the Institute of Engineering Mechanics, China Earthquake Administration and China Earthquake Networks Center, National Earthquake Data Center (http://data.earthquake.cn, accessed on 31 October 2021). The sampling rate is 200 Hz. The rupture duration of small earthquakes is short, and P waves and S waves are usually separated in time. Large earthquakes rupture for a long time and P waves are continuously emitted during the rupture, which results in P waves and S waves sometimes partly overlapping. It means that it is difficult to detect P waves. However, the large earthquake rupture growth is large enough, although the initial S waves make some interference, and P waves can still be clearly detected [23].

Figure 6 shows the records of U−D direction of the M7.0 earthquake and the corresponding results of STP/LTP, STP, LTP calculation. It can be seen that the waveform of the large earthquake lasts for a long time. In Figure 6c,d, STP/LTP value will be at a high level within a period of time after the arrival of the P waves; $P_{SNR}$ value is also determined in a very short time. In Figure 6e, the relative power variation in short-time window and long-time window of large earthquake is the same as that of small earthquake. Different from small earthquakes, the relative power value of the seismic signal in short-time window and long-time window of large earthquakes is higher than that of small earthquakes, and the duration is longer than that of small earthquakes, which also reflects the characteristics of large earthquakes.

Compared with small earthquakes, the energy of large earthquakes is very strong, and the duration of earthquakes is longer. According to the rule of small earthquakes, the large earthquake response also conforms to the relationship of energy attenuation with hypocentral distance. Figure 7 shows the relationship between the energy of the earthquake and the hypocentral distance in the M7.0 earthquake. It can be seen that when the hypocentral distance reaches 100 km, the $P_{SNR}$ value also drops to about 6, and when the distance is larger, the $P_{SNR}$ value will also be submerged by noise. Different from small earthquakes, for the same hypocentral distance, the $P_{SNR}$ of large earthquakes is significantly higher than that of small earthquakes, and the attenuation of the $P_{SNR}$ in the large earthquakes is slower, which indicates the characteristics of the large earthquake energy. In addition, it should be noted that for the large earthquakes, the hypocentral distance of the stations should not be too small. When the station is close to the seismic center, on the one hand, it is difficult to distinguish between P waves and S waves, which affects the detection of P waves. On the other hand, due to the overlap of P waves and S waves, the $P_{SNR}$ value is too large, which is not within the range of $P_{SNR}$ attenuation relationship.

### 3.3. Comprehensive Analysis

For the earthquakes with different magnitude ranges, Figure 8 shows the distribution of $P_{SNR}$ value on hypocentral distance *R*. The earthquakes ranging from M3.0 to M4.0, M4.5 to M6.0, and large earthquakes ranging from M7.0 to M8.0, are selected for comparative analysis. It can be seen that for the same hypocentral distance, the $P_{SNR}$ value of large earthquakes is significantly higher than that of small earthquakes. With the increase in the magnitude, the energy increases, and the $P_{SNR}$ value will also increase. In the same magnitude range, the $P_{SNR}$ value has also a certain attenuation trend with the hypocentral distance. In each magnitude range, the $P_{SNR}$ values are distributed according to a certain linear relationship. There are also some discrete points in the diagram, and even some discrete points are located in other magnitude ranges, which is realistic and indicates the complexity of the process of the seismic energy release.

A large number of studies have shown that when the earthquake energy reaches a certain degree, the magnitude saturation phenomenon will occur. By comparing the M7.0 earthquake and M8.0 earthquake in Figure 8, it is found that $P_{SNR}$ has no stratification, which also indicates the magnitude saturation of $P_{SNR}$. But the specific saturation magnitude needs the further study to determine.

In order to study the stability and accuracy of the $P_{SNR}$ method, the $P_{d}$ method is chosen to compare with it. The fitting models of the $P_{d}$ method and $P_{SNR}$ method are shown in Equations (10) and (11). Among them, the arrival time of P waves is extracted by the manual supervision method, the short window length is set as 0.3 s, and the long window length is set as 3 s. In the $P_{d}$ method, the high pass filter for integrating displacement is set as the second-order high pass Butterworth filter (the low-frequency cut-off frequency is 0.025 Hz), since the $P_{d}$ value determined by the first 3 s P waves cannot be used for *M* ≥ 6.7 earthquakes [13], and considering the accuracy of the algorithm, the time window for calculating $P_{d}$ value is set as 4 s.
(10)$$M=A\times log\left(P_{d}\right)+B\times log\left(R\right)+C\;$$
(11)$$M=A\times log\left(P_{SNR}\right)+B\times log\left(R\right)+C\;$$
where *A*, *B*, and *C* are the coefficients to be fitted.

The study of Zollo shows that the $P_{d}$ method also has a saturation phenomenon [24]. When the time window length is 4 s, the saturation magnitude can reach M7.0. Therefore, the *M* ≤ 7.0 earthquakes in the samples are compared and analyzed by $P_{d}$ method and $P_{SNR}$ method in this study, as shown in Table 1. The fitting equation by $P_{SNR}$ method is *M* = −4.6912 + 4.2519log$P_{SNR}$ + 3.8137logR, and the fitting correlation coefficient reaches 0.8012, which conforms to the linear relationship. The fitting equation by $P_{d}$ method is *M* = −1.2729 + 1.3405log$P_{d}$ + 5.4202log*R*, and the fitting coefficient is 0.7552, which conforms to the linear relationship. As can be seen from Table 1, the $R^{2}$ statistic of $P_{SNR}$ is slightly larger than that of $P_{d}$, and the mean residual of the two methods are also roughly the same, which represents the high accuracy and stability of this method. Figure 9 shows the residual diagram of the two methods. From the discreteness of the residuals, the residuals of the two methods are distributed around 0, with only a few outliers. It can be seen that the fitting results of the two methods are good.

After the fitting model is obtained, the accuracy of the model is tested by detecting several groups of seismic events. Table 2 shows the test results. *N* is the number of stations involved in each event. $M_{a}$ is the actual magnitude. $M_{e}$ is the estimated magnitude. In each group of earthquake events, the $P_{SNR}$ value and hypocentral distance obtained by effective stations are brought into the model, and the magnitude estimation results of each station are obtained. Finally, the average value is the magnitude estimation result of the earthquake event, which compensates for possible site effection. Each seismic event can roughly estimate the magnitude of the earthquake through only several stations, and these effective stations are close to the source, which also makes the magnitude quickly obtained through close stations when the earthquake occurs.

Figure 10 shows the variation of the estimated magnitude $M_{e}$ with the number of the stations *N* in the xingwen M5.7 event. With the increase in the *N*, the Me estimated by $P_{SNR}$ is more and more equal to the actual magnitude $M_{a}$. In the EEW with the dense sensor network, it is possible to obtain the more accurate magnitude by extracting the characteristic parameter $P_{SNR}$ from the multiple sensors for estimation and averaging.

In terms of timeliness, the time of $P_{SNR}$ in all sample data are counted, as shown in Figure 11. Each interval of the abscissa in the histogram is the time range after the arrival of P waves. Most of the time corresponding to the $P_{SNR}$ ranges from 0–0.3 s, which is because the length of the short-time windows is 0.3 s, and there is no additional P waves’ power in the long-time windows. In this time range, the $P_{SNR}$ is most likely to appear, which indicates the strong timeliness of the $P_{SNR}$ method. A few times range from 0.3–2 s. It can be determined that the occurrence time of $P_{SNR}$ value is within 2 s after the arrival of P waves, but the concrete time is still not certain. The process of fault rupture is very complex. The energy of the small earthquakes is small, and the duration is short. The energy of the large earthquakes is strong, and the duration of seismic waves is long. Even for the same earthquake, due to the inherent noise of the instrument and the local geological conditions, the determination time of $P_{SNR}$ value is slightly different. The $P_{SNR}$ method has better timeliness, which can greatly save costs and greatly reduce the time of magnitude estimation.

Figure 12 shows two examples of $P_{SNR}$ values in 0–0.3 s and 1.5–1.8 s after P waves’ arrival, the arrival time of P waves, $P_{SNR}$ and $P_{d}$ are also shown in the figure. In the first example (a, b, c, d), after the arrival of P waves, the STP/LTP value immediately increased to the maximum, and the $P_{SNR}$ value quickly appeared. In the second example (e–h), the STP/LTP value increases after the arrival of P waves, but it increases to the maximum after a period of time. Although the occurrence time of $P_{SNR}$ is different between the two examples, it is much earlier than the $P_{d}$ method. With the comprehensive consideration of timeliness and accuracy, the $P_{SNR}$ method has significant advantages. The magnitude estimation can be carried out synchronously when P waves arrives, which greatly improves the timeliness of earthquake early warning system. However, $P_{SNR}$ requires high accuracy for P waves’ arrival extraction. If P waves’ seismic phase identification is wrong, this method will also result in a false alarm.

After analyzing the accuracy and timeliness of *P_SNR_* algorithm, it can be seen that $P_{SNR}$ can synchronize the extraction of P waves’ arrival time for magnitude estimation in the initial short time of P waves. *P_SNR_* is suitable for the construction of earthquake early warning system in terms of timeliness, accuracy, and robustness.

## 4. Discussion

The fast magnitude estimation method (*P_SNR_*) based on SNR feature proposed in this paper is a new method in the field of magnitude estimation. The significance of this study is that the *P_SNR_* method can quickly estimate the magnitude in a short time; in particular, the magnitude estimation time is controlled within two seconds after the arrival of the P waves, which greatly meets the timeliness of the earthquake early warning system. Compared with the existing magnitude estimation methods, the new method greatly reduces the time used for magnitude estimation and improves the efficiency of earthquake early warning, which is the main highlight of the *P_SNR_* method.

The *P_SNR_* method extends the advantages of the STP/LTP method and extends from the time-of-arrival extraction of P waves to the magnitude estimation, so that the magnitude estimation can be synchronized with the extraction of P waves, which is the cleverness of the *P_SNR_* method.

With the development of sensor technology and computer technology, the network of seismic monitoring stations will be more and more intensive. Under such technical background, the efficiency of earthquake early warning will be significantly improved. Fortunately, the calculation cost of *P_SNR_* method is low. After the P waves are extracted by STP/LTP method, it is only necessary to find the *P_SNR_* value, without excessive calculation, which can effectively reduce the pressure of the single chip microcomputer at the equipment end. Based on the mode of cloud-edge collaboration, in the earthquake early warning system, the edge equipment can directly record the *P_SNR_* value and transmit it to the cloud server, which also greatly reduces the transmission pressure of the network and the calculation pressure of the cloud server.

The *P_SNR_* method is more robust than the existing methods, which can be applied to weak noise, general noise environment, and is not easily affected by instrument background noise. The *P_SNR_* parameter performance is very stable in a weak noise environment. Moreover, the *P_SNR_* method has the advantage of dimensionless, which can be widely used in different types of station instruments and improving the efficiency of data processing. *P_SNR_* can be obtained directly from the measuring units of the instruments used for different specifications. In the station data of this study, the instrument specifications are not completely uniform, the use of the *P_SNR_* method greatly saves the cost of data processing and avoid the calculation error caused by different units.

It should be noted that the geological structure and crustal movement are different in different regions. All earthquake events in this paper are shallow earthquakes. When the $P_{SNR}$ parameter is used to estimate the magnitude, local historical records should be used for model training so as to improve the accuracy of the model and reduce the false alarm rate.

In addition, the factors affecting the $P_{SNR}$ value cannot be ignored. The premise of obtaining *P_SNR_* is to detect the P waves accurately. Many research methods have achieved good results in P waves’ detection, such as the STP/LTP method [21], the AIC method [25], and the deep learning method [26]. The length of the long-time window and short-time window in sliding window also has a certain influence on *P_SNR_* value. If the length of the short-time window is too short, there will be noise interference. If the length of the short time window is too long, the seismic effective energy cannot be expressed in time. Finally, the more stable the sensor performance for seismic monitoring is, the more accurate and stable the *P_SNR_* value and the accuracy of the fitting model are.

## 5. Conclusions

Aiming at the timeliness and accuracy of a magnitude estimation in an earthquake early warning system through analyzing the energy characteristics of the STP/LTP method and combining the principle of SNR and the definition of relative power, a P waves arrival fast magnitude estimation method (*P_SNR_*) based on SNR is proposed in this paper. The main conclusions are as follows:(1)Both ghd *P_SNR_* method and *P_d_* method have high stability and accuracy, and the *P_SNR_* method has strong robustness.(2)The highlight is that the *P_SNR_* method has good timeliness. The time corresponding to *P_SNR_* value is usually within 2 s from the first arrival of P waves, and sometimes it can even be synchronized to the first arrival of P waves, which can greatly save time in the earthquake early warning magnitude estimation. *P_SNR_* is a good choice that considers both timeliness and accuracy.(3)The STP/LTP method based on SNR and relative power contains certain energy information when seismic P waves arrive. There is a certain positive correlation between *P_SNR_* value and magnitude, and there is an obvious attenuation relationship between *P_SNR_* value and hypocentral distance *R*.(4)Owing to the dimensionless characteristics, the *P_SNR_* method can be widely used in different sensors and reduce costs in the processing of seismic records.

In all, *P_SNR_* method can meet the needs of the EEW in terms of timeliness and accuracy.

## Figures and Tables

**Figure 1 sensors-22-04534-f001:**
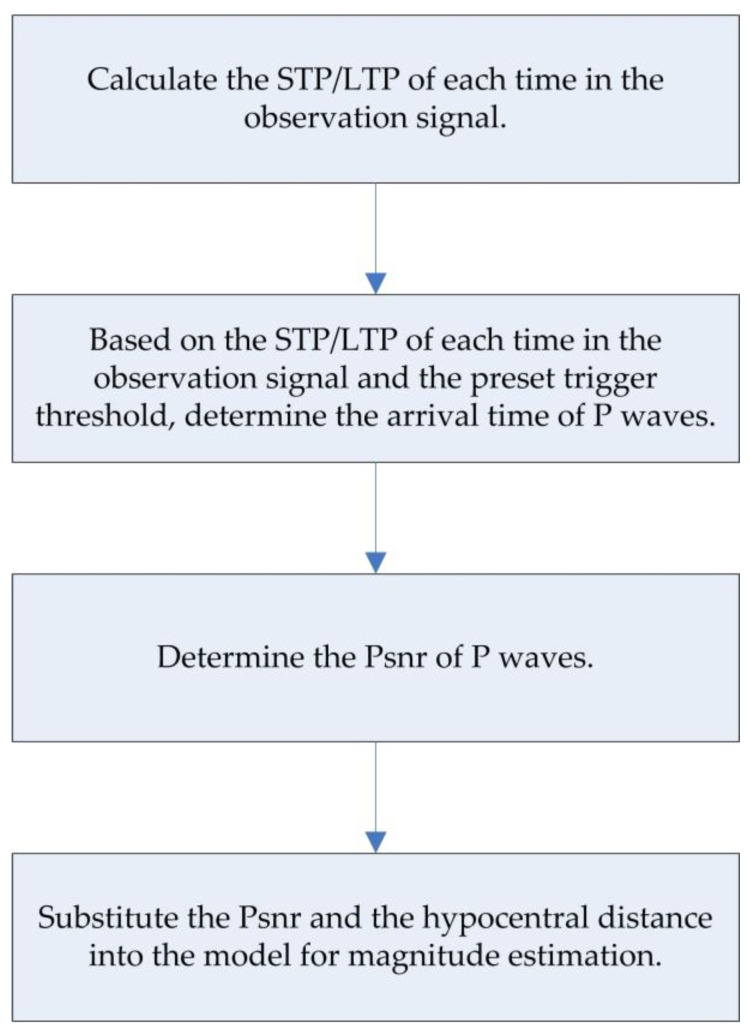
Application flowchart of the new magnitude estimation method (*P_SNR_*).

**Figure 2 sensors-22-04534-f002:**
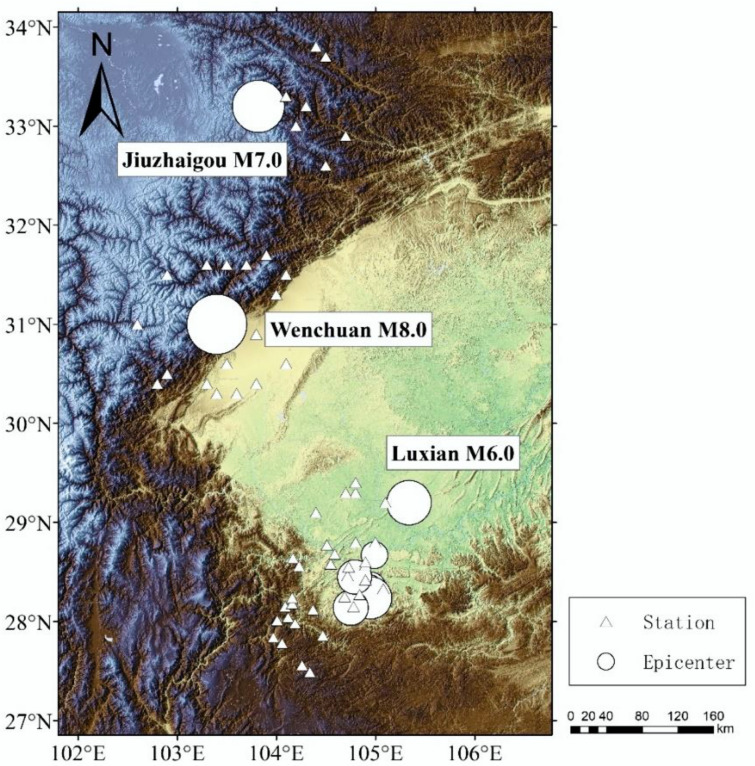
Locations and distributions of the main epicenters and stations. For the *M* ≤ 5.0 earthquakes, the hypocentral distance is within 50 km, and for the *M* > 5.0 earthquakes, the hypocentral distance is within 100 km.

**Figure 3 sensors-22-04534-f003:**
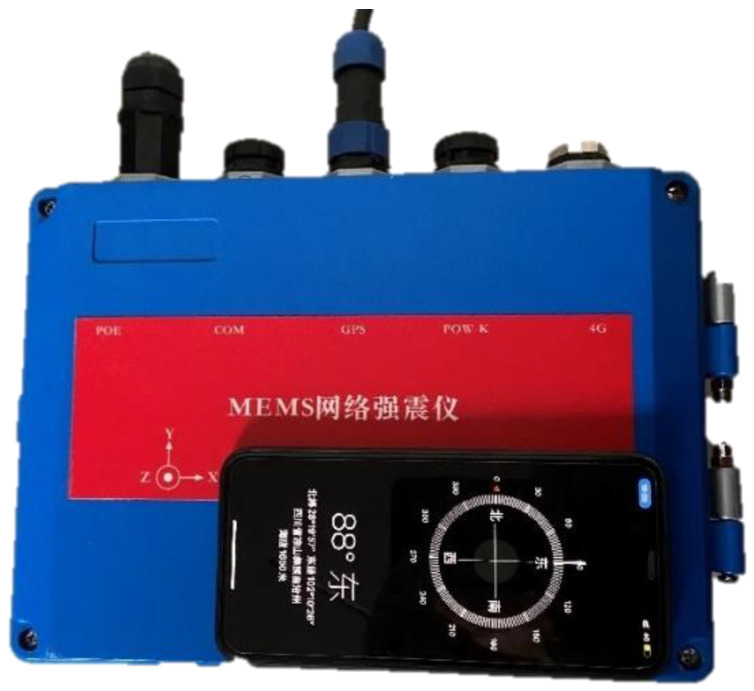
The photograph of the MNSMS. The text in the red label is MEMS Network Strong Motion Seismograph. The text on screen is the coordinate.

**Figure 4 sensors-22-04534-f004:**
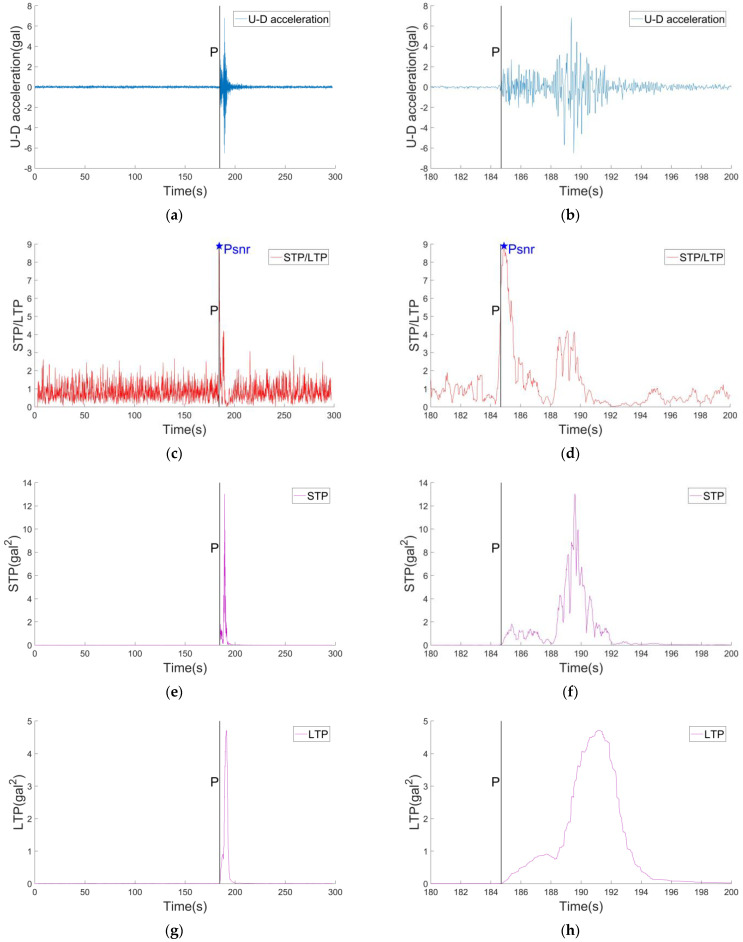
(**a**) The record of the U−D direction in the M4.0 earthquake. (**b**) The local enlargement result of (**a**). (**c**) The STP/LTP result of the record. The length of the short-time window is 0.3 s. The length of the long-time window is 3 s. (**d**) The local enlargement result of (**c**). (**e**) The STP result of the record. (**f**) The local enlargement result of (**e**). (**g**) The LTP result of the record. (**h**) The local enlargement result of (**g**).

**Figure 5 sensors-22-04534-f005:**
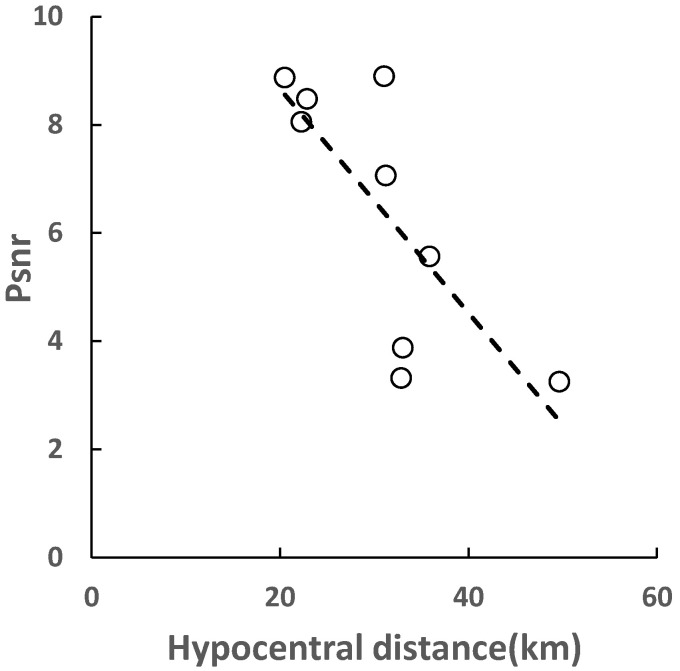
The relationship between and hypocentral distance in the M4.0 earthquake.

**Figure 6 sensors-22-04534-f006:**
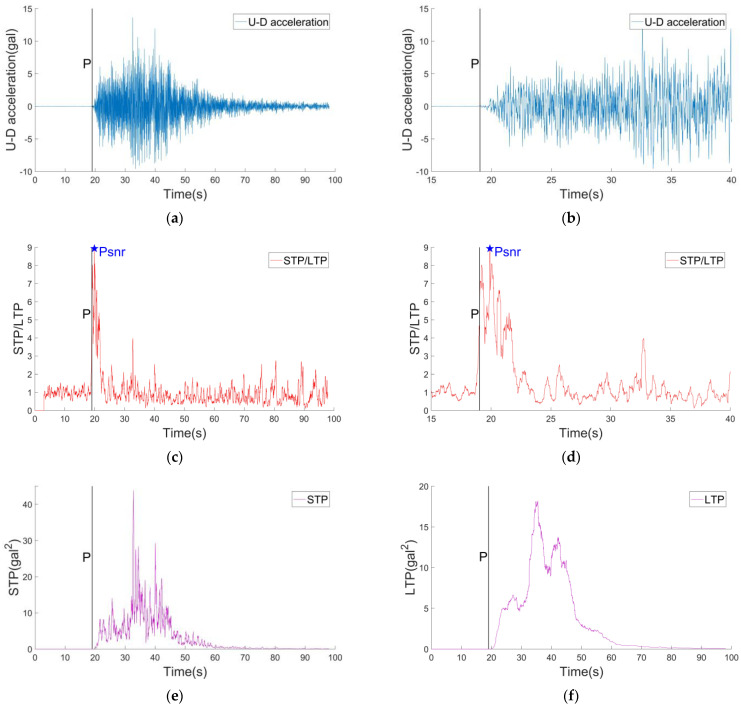
(**a**) The record of the U−D direction in the M7.0 earthquake. (**b**) The local enlargement result of (**a**). (**c**) The STP/LTP result of the record. The length of the short-time window is 0.3 s. The length of the long-time window is 3 s. (**d**) The local enlargement result of (**c**). (**e**) The STP result of the record. (**f**) The LTP result of the record.

**Figure 7 sensors-22-04534-f007:**
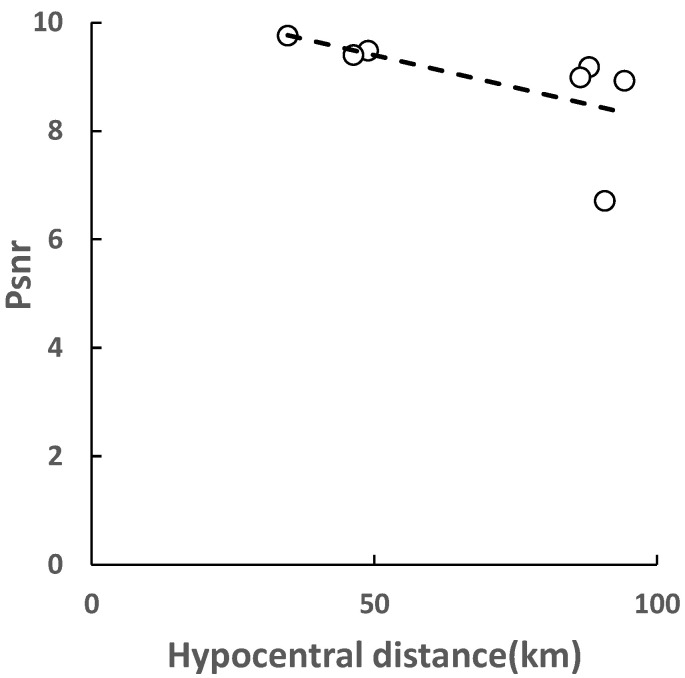
The relationship between *P_SNR_* and hypocentral distance in the M7.0 earthquake.

**Figure 8 sensors-22-04534-f008:**
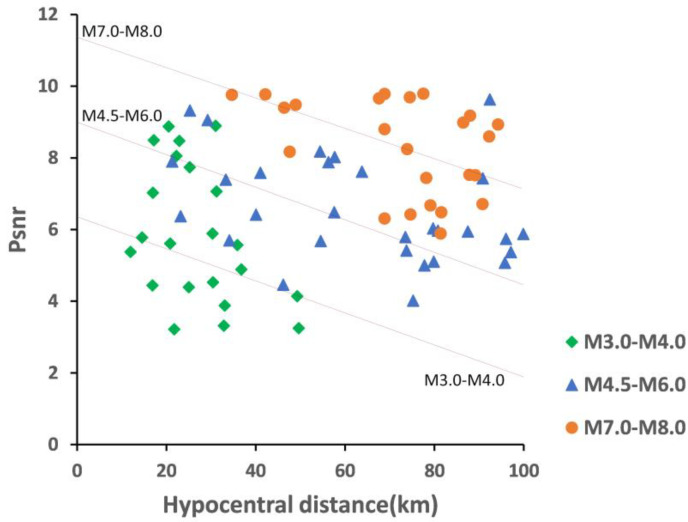
Attenuation relationship between $P_{SNR}$ and hypocentral distance R. The green points in the figure are the $P_{SNR}$ values of magnitude M3.0 to M4.0; the blue points are the $P_{SNR}$ values of magnitude M4.5 to M6.0; the orange points are the $P_{SNR}$ values of magnitude M7.0 to M8.0.

**Figure 9 sensors-22-04534-f009:**
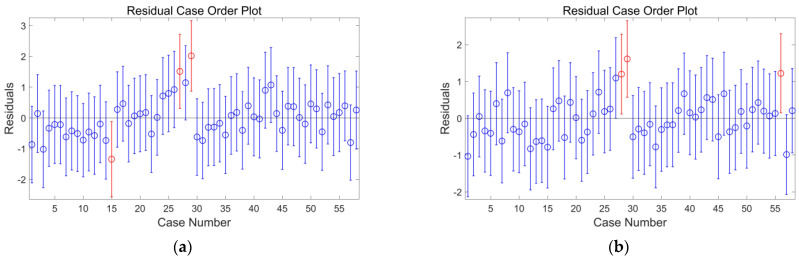
(**a**,**b**) are residual graphs of *P_d_* method and *P_SNR_* method, respectively. The blue points are the residual points in the normal range, and the red points are the abnormal points.

**Figure 10 sensors-22-04534-f010:**
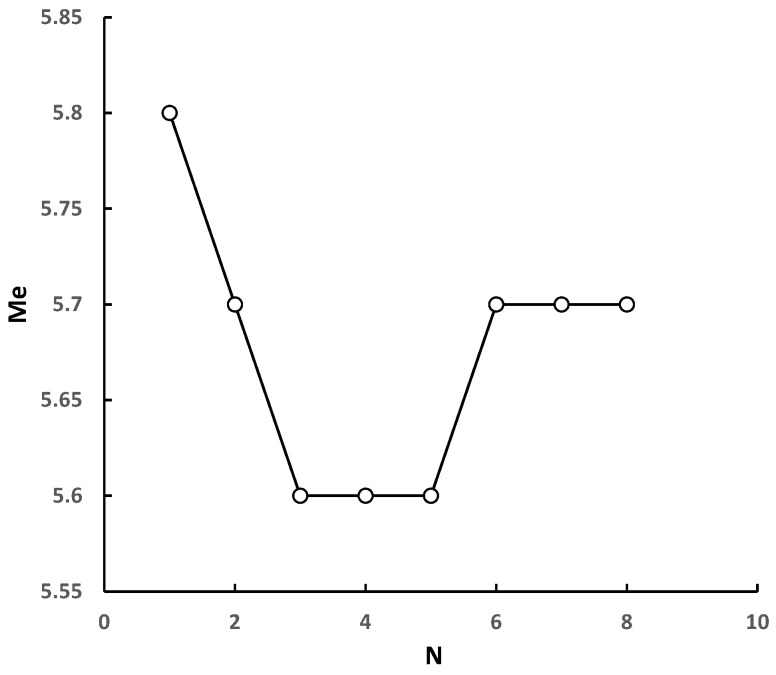
The variation of the *Me* with the number of the stations *N* in the Xingwen M5.7 event.

**Figure 11 sensors-22-04534-f011:**
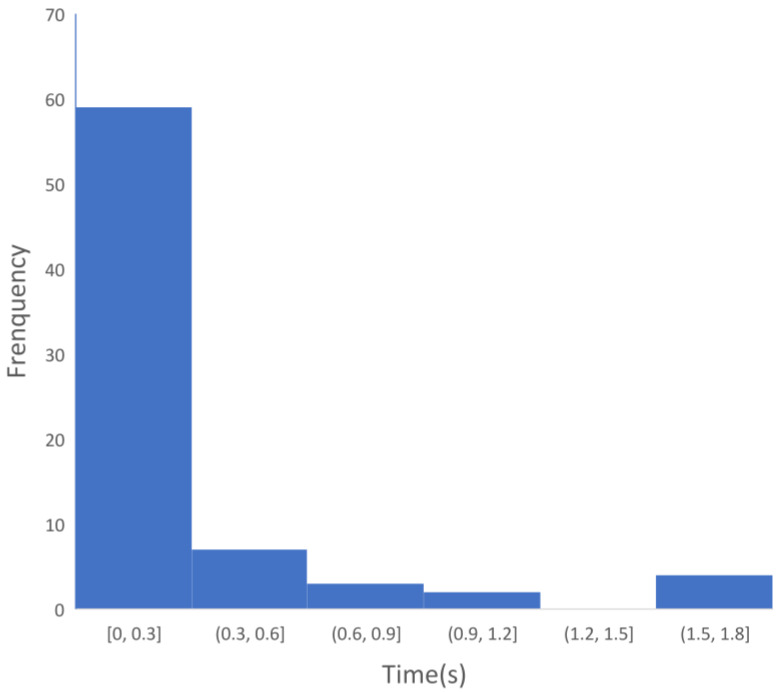
Time-consuming distribution histogram of *P_SNR_*.

**Figure 12 sensors-22-04534-f012:**
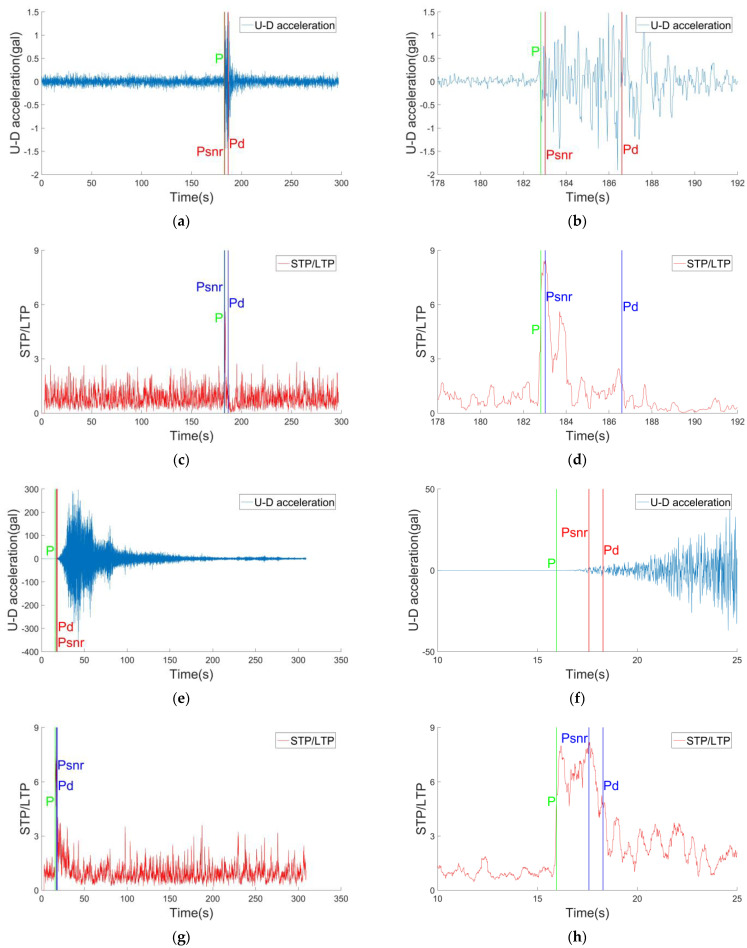
(**a**–**d**) are the results of the Timeliness comparison of *P_SNR_* and *P_d_* in the M4.0 earthquake. (**e**–**h**) are the results of the Timeliness comparison of *P_SNR_* and *P_d_* in the M8.0 earthquake. (**b**,**d**,**f**,**h**) are local enlargement results of figures (**a**,**c**,**e**,**g**), respectively. The green lines in all figures represent the arrival time of the P wave. The red lines in (**a**,**b**,**e**,**f**) represent the time of the *P_SNR_* and *P_d_*, respectively. The blue lines in (**c**,**d**,**g**,**h**) represent the time of the *P_SNR_* and *P_d_*, respectively.

**Table 1 sensors-22-04534-t001:** Comparisons of *P_d_* and *P_SNR_* methods in all samples.

	*P_d_*	*P_SNR_*
*R* ^2^	0.7552	0.8012
Variance of error	0.4074	0.3308

**Table 2 sensors-22-04534-t002:** Test results of the *P_SNR_* method.

Event, Date	*N*	$M_{a}$	$M_{e}$
Gongxian, 13 March 2021	2	3.0	3.4
Gongxian, 7 November 2020	3	3.4	3.7
Gongxian, 23 June 2020	2	3.8	3.2
Gongxian, 12 June 2020	2	4.0	3.7
Gongxian, 15 February 2021	2	4.0	4.3
Gongxian, 13 November 2020	2	4.1	4.2
Xingwen, 16 December 2018	8	5.7	5.7
Luxian, 16 September 2021	2	6.0	5.5

## Data Availability

The data for this study were provided by the National Key Research and Development Program of China (No. 2019YFC1509205), Institute of Engineering Mechanics, China Earthquake Administration and China Earthquake Networks Center, National Earthquake Data Center (http://data.earthquake.cn, accessed on 31 October 2021).

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
