# Peer review of "A Synchronous Magnitude Estimation with P-Wave Phases’ Detection Used in Earthquake Early Warning System"

_sensors, 2022, doi:10.3390/s22124534_

Round 1
Reviewer 1 Report
This is an interesting manuscript using the short-term power and long-term power ratio (STP/LTP) to determine the P arrival as well as magnitude. My comments as follows,
Major comment
The STP/LTP is a relative amplitude parameter. It does not represent the absolute amplitude or energy. The background signals (LTP) significant influence the ratio of STP/LTP. In principle, it includes technical error in magnitude determination.
Minor comments
1. STP/LTP is one of short term average over long term average (STA/LTA) with different characteristic function. It will be good use STA/LTA to instead STP/LTP.
2. A lot of localities appears in text. It will be nice to show in figure.
Reviewer 2 Report
Major notes:
1. The goal of the study is to provide a fast and accurate magnitude estimation for an EEW application. While the method is described in detail, important test results are missing. In my opinion, it is important to evaluate the behavior of the magnitude estimate in the process of incrementing measurements. This is what will happen in real EEW systems. It is important to understand whether the estimate becomes more stable on average as new station measurements become available. I suggest that the authors provide one or two plots showing the behavior of the estimate as N increases, for example from 2 to 10 for several earthquakes. It is also important to show how the actual magnitude converges through the standard routine on the same plot.
2. When introducing a new method for estimating magnitude, cross-magnitude relations should be presented in a clear manner. For example, the Psnr vs. Mw. This allows the reader to better perceive the new approach.
Minor notes:
1. It should be explained what U-D direction is.
2. In my opinion, the text on lines 207-223 should be reformulated to make it more compact and clear.
3. I'm not sure that all earthquakes with M<5.0 can be attributed to micro-earthquakes (line 265). I propose to use more convenient terminology from seismology.
4. I think it would be better to provide the sample rate in Hz rather than time interval in seconds (line 280).
Round 2
Reviewer 1 Report
The STP/LTP is relatively amplitude parameters. It exists a technical problem on magnitude determination. I suggest that use STP as the parameter to determine magnitude. It can solve this problem and result will be same.
